# Atomic-level structure determination of amorphous molecular solids by NMR

Manuel Cordova [1,2], Pinelopi Moutzouri[1], Sten O. Nilsson Lill[3], Alexander Cousen[4], Martin Kearns[5], Stefan T. Norberg [6], Anna Svensk Ankarberg [6], James McCabe[5], Arthur C. Pinon[7], Staffan Schantz [6] ✉ & Lyndon Emsley [1,2] ✉

Structure determination of amorphous materials remains challenging, owing to the disorder inherent to these materials. Nuclear magnetic resonance (NMR) powder crystallography is a powerful method to determine the structure of molecular solids, but disorder leads to a high degree of overlap between measured signals, and prevents the unambiguous identification of a single modeled periodic structure as representative of the whole material. Here, we determine the atomic-level ensemble structure of the amorphous form of the drug AZD4625 by combining solid-state NMR experiments with molecular dynamics (MD) simulations and machine-learned chemical shifts. By considering the combined shifts of all $^1H$ and $^{13}C$ atomic sites in the molecule, we determine the structure of the amorphous form by identifying an ensemble of local molecular environments that are in agreement with experiment. We then extract and analyze preferred conformations and intermolecular interactions in the amorphous sample in terms of the stabilization of the amorphous form of the drug.

Structure-activity relations drive most areas of modern chemistry. For example, the design of efficient and safe pharmaceutical drugs can be rationalized through the understanding of their atomic-level structure. This can greatly accelerate the search for new compounds with specific properties[1–3]. Tools to determine atomic-level structures have thus become a vital part of modern chemistry research. This is a particular challenge for powdered molecular solids.

In contrast to methods such as powder X-ray diffraction[4–6] or electron diffraction[7–12], NMR directly probes the local atomic environment, allowing for structural characterization without the need for long-range order[13]. In this direction, solid-state NMR has seen spectacular progress in the last few years[13–16], and methods have been introduced to solve crystal structures of bulk inorganic[17–20] or

molecular solids[14,15,21–28]. This has resulted in successful structure determination of a variety of powdered materials[13], including organic solids[14,23–25,29–34], enzyme active sites[35], cementitious materials[36–38], battery materials[39,40], and hybrid perovskite materials[41]. These structures have been solved by comparing density functional theory (DFT) chemical shifts (or other NMR parameters) computed on model structures (typically generated through crystal structure prediction (CSP) protocols) with experimental values[22–25].

Despite these remarkable results, complete atomic-level structure determination of amorphous molecular solids remains extremely challenging[42,43]. Nevertheless, amorphous solids are becoming increasingly important. For example, the development of amorphous drug formulations is of current high interest in the pharmaceutical

[1]Institut des Sciences et Ingénierie Chimiques, École Polytechnique Fédérale de Lausanne (EPFL), CH-1015 Lausanne, Switzerland. [2]National Centre for Computational Design and Discovery of Novel Materials MARVEL, École Polytechnique Fédérale de Lausanne (EPFL), Lausanne, Switzerland. [3]Data Science & Modelling, Pharmaceutical Sciences, R&D, AstraZeneca, Gothenburg, Sweden. [4]Early Chemical Development, Pharmaceutical Sciences, R&D, AstraZeneca, Macclesfield, UK. [5]Early Product Development and Manufacturing, Pharmaceutical Sciences, R&D, AstraZeneca, Macclesfield, UK. [6]Oral Product Development, Pharmaceutical Technology & Development, Operations, AstraZeneca, Gothenburg, Sweden. [7]Swedish NMR Center, Department of Chemistry and Molecular Biology, University of Gothenburg, 41390 Gothenburg, Sweden. ✉e-mail: Staffan.Schantz@astrazeneca.com; lyndon.emsley@epfl.ch

industry, owing to their enhanced solubility and bioavailability with respect to crystalline drugs[44–47]. However, in the absence of methods for atomic-level structure determination, it is not possible to rationalize the factors that lead to the stabilization of amorphous forms, which is a crucial step in developing stable formulations.

The disorder inherent to these compounds leads to the broadening of NMR signals, which leads to significant overlap between the peaks associated with different atomic sites. Consequently, this increases the need for multi-dimensional experiments, which are more difficult to obtain than for crystalline materials due to the lower sensitivity associated with broader lineshapes. The assignment of chemical shifts for amorphous compounds is thus often challenging. Recent advances in dynamic nuclear polarization (DNP)[48–50] have resulted in significant gains of sensitivity in crystalline and amorphous molecular solids, leading to a significant reduction in experimental time required to obtain multi-dimensional NMR spectra of solids.

In addition to these experimental considerations, modeling amorphous structures of materials generally requires the use of molecular dynamics (MD) simulations of large cells typically containing hundreds of molecules. This results in a prohibitive cost for computing chemical shifts using DFT for such large systems. Several approaches have been introduced in order to circumvent this drawback, ranging from using small (hundreds of atoms) amorphous system sizes[36–38,51,52] to isolating local environments to compute chemical shift[35,53,54] to including the effect of long-range interactions by approximate methods[55–59]. While these methods do enable the computation of chemical shifts at the DFT level of theory for amorphous solids, the computational cost remains significant, preventing large-scale chemical shift computations.

Structural disorder has been investigated in proteins by a combination of solid-state NMR, structure generation algorithms, and chemical shift predictions[60–62]. However, such studies have relied on models of chemical shifts in proteins based in part on their primary and/or secondary structure[63–66]. Such models are thus not directly applicable to other molecular solids.

Machine learning (ML) models developed in recent years have proven able to reproduce quantum mechanical properties of materials with similar accuracy as DFT, and at a fraction of the computational cost[67,68]. In particular, ML models of chemical shifts have been introduced and shown to be as accurate as DFT[64,65,69–78]. Recently, we introduced ShiftML, an ML model trained to predict chemical shifts in molecular solids[79]. In its most recent version, the model is trained to reproduce DFT results for solids containing up to 12 elements, and includes distorted geometries, which would be key to describing amorphous systems[80].

We previously showed how combining MD simulations with large-scale chemical shift predictions obtained using ShiftML allowed the determination of the hydrogen bonding structures in an amorphous drug by comparison with experimentally obtained shifts[42]. However, this approach used a single chemical shift, to focus on the determination of the hydrogen bonding motifs in the structure, as a proof of concept.

Here, we determine the complete ensemble atomic-level structure of the amorphous drug AZD4625[81,82] through the combination of DNP-enhanced solid-state NMR, MD, and machine-learned chemical shifts. To do this we introduce a general approach that integrates multiple chemical shifts and includes the experimental spread of chemical shift distributions in NMR spectra of molecular solids, that we use to select an ensemble of local molecular environments that best match the chemical shift distributions in the measured spectra. This process is applied to over one million molecules from MD simulations, for which we predict chemical shifts. From an analysis of the extracted ensemble of local molecular environments in best agreement with the experiments, we identify key intermolecular interactions and conformations present in the amorphous sample. The local atomic

environments determined by NMR were found to accurately reproduce the radial distribution function measured for the sample by powder X-ray diffraction, and to correspond to energetically favorable local structures.

## Results

Figure 1 shows the chemical structure of AZD4625 and the labeling scheme used here, as well as the experimental 1D and 2D NMR spectra obtained for the amorphous form of AZD4625. The spectra display broad linewidths, typical of disordered systems. This highlights the need for multi-dimensional experiments in order to obtain a confident assignment, by spreading the signals over multiple dimensions. With this set of spectra the $^1$H and $^{13}$C chemical shifts obtained were assigned as described in the Methods section, leading to the assignments given in Supplementary Table 6. By fitting Gaussian functions to resolved peaks in the 1D $^1$H and $^{13}$C MAS spectra, and 2D $^1$H-$^1$H DQ/SQ spectrum, we obtained linewidths between 2 and 6 ppm for $^{13}$C, 0.6 and 1 ppm for C–H protons, and 1.8 ppm for the OH proton (see Supplementary Table 6 and Supplementary Figs. 2–5). Here, we assume Gaussian shapes for all experimental distributions of chemical shifts. The extracted experimental chemical shift distributions will then serve as the basis to score molecular environments as described in the Methods section. We note that no crystalline form of pure AZD4625 has previously been reported.

To generate a broad ensemble of possible structures, eight MD simulations were carried out with cells containing 128 molecules of AZD4625, randomly initialized in order to model the amorphous system, as described in the Methods section. Chemical shift predictions performed using ShiftML2 were then compared with the experimental values obtained for $^1$H and $^{13}$C (excluding the protons and carbon labeled 1 in Fig. 1a due to the ambiguity in their assignment). A total of 1,025,280 molecular environments, each comprising a central molecule and all molecules that have at least one atom within 7 Å from any atom of the central molecule (see Methods section), were extracted from the MD snapshots. For each atomic site in the central molecule of a molecular environment, we compute the probability that the predicted shift is drawn from the corresponding experimental chemical shift distribution. The probabilities across all atomic sites are then combined into a global probability that the local molecular environment matches the NMR experiments. More details are given in the Methods section. Fig. 2a shows the root-mean-square error (RMSE) between $^1$H and $^{13}$C chemical shifts computed for all AZD4625 molecules in each of the 8010 snapshots taken from the MD trajectories, as well as the calculated probability that the local molecular environment of each molecule is consistent with the NMR experiments. This includes the computation of chemical shifts for over a million molecules. As expected, higher probability is correlated with lower $^1$H and $^{13}$C shift RMSE, but it is very important to note that the RMSEs only consider the difference between the center of the experimental distributions of shifts, and the corresponding chemical shift prediction for each atomic site, while the probability calculated using Eqs. 1–4 also take into account the width of the experimental distributions as well as the prediction uncertainty, providing an improved picture of the compatibility of a given local molecular environment with the experiments. The histogram of all probabilities of local molecular environments ($p_j$) to match the experiments is shown in Fig. 2b. Here, we selected the 1% of local molecular environments in best agreement with the experiment to construct the NMR ensemble, which corresponds to the probabilities above 33%, as indicated by the dashed vertical line in Fig. 2b.

Here, we independently select molecular environments compatible with the NMR experiments. The generation of environments through the MD simulations is inherently biased by the force field used and the starting configurations. The selection of the subset that best matches the experimental data does not aim here to reproduce the

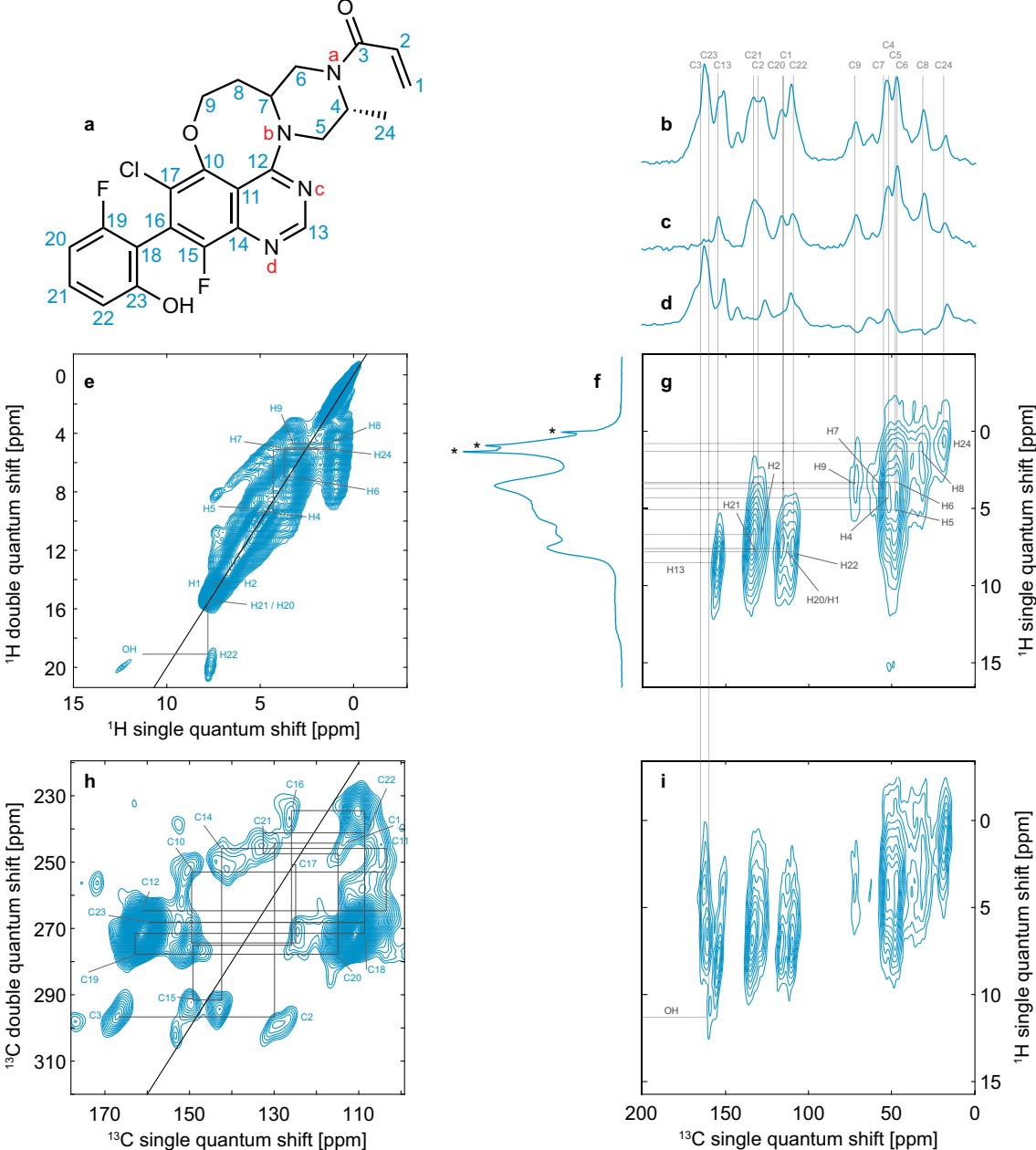

**Fig. 1 | NMR spectra of the amorphous form of AZD462 used for chemical shift assignment. a** Chemical structure of AZD4625 and carbon (blue numbers) and nitrogen (red letters) labeling schemes used here. 1D (**b**–**d**) DNP-enhanced $^{13}$C CPMAS spectra without (**b, c**) and with (**d**) CPPI spectral editing. **f** 1D $^1$H 100 kHz MAS spectrum. 2D (**e**) $^1$H-$^1$H DQ/SQ, (**g, i**) DNP-enhanced $^1$H-$^{13}$C DUMBO-HETCOR and (**h**) DNP-enhanced $^{13}$C-$^{13}$C INADEQUATE spectra of amorphous AZD4625. In **d** -CH$_2$ groups appear negative, -CH groups disappear, and -C and -CH$_3$ groups retain a positive intensity. The gray lines indicate correlated peaks or $^{13}$C chemical shifts of protonated carbon species. The stars in **f** indicate artifacts due to mobile impurities in the rotor.

exact experimental ensemble of molecular environments in the sample (as is done, e.g., in NMR studies of intrinsically disordered proteins[83–85]), but here it provides an additional bias in order to identify systematic structural differences from the ensemble generated by MD, as seen below.

Figure 2c–e shows the histograms of chemical shifts computed for carbon labeled 3, proton labeled 13, and of the OH proton for all AZD4625 molecules in the MD trajectories as compared to those from the NMR ensemble. These examples are taken to illustrate the typical changes of chemical shift distributions seen upon the selection of local atomic environments. The distributions for all other protons and carbons considered are given in Supplementary Figs. 7–11. The

distribution of predicted shifts for carbon labeled 3 (Fig. 2c) was found to be significantly closer to the experimental distribution of shifts upon selection of local molecular environments, suggesting that this chemical shift does discriminate between the structures. In contrast, for example, the distribution of predicted shifts for the proton labeled 13 (Fig. 2d), which already displays a large overlap with the corresponding experimental distribution of shifts, does not display a significant change upon the selection of local molecular environments. Then we note that the distribution of predicted chemical shifts for the OH proton (Fig. 2e) displays a large difference after the selection of local molecular environments, again suggesting that this shift is a powerful discriminator. However, even after the selection of the best

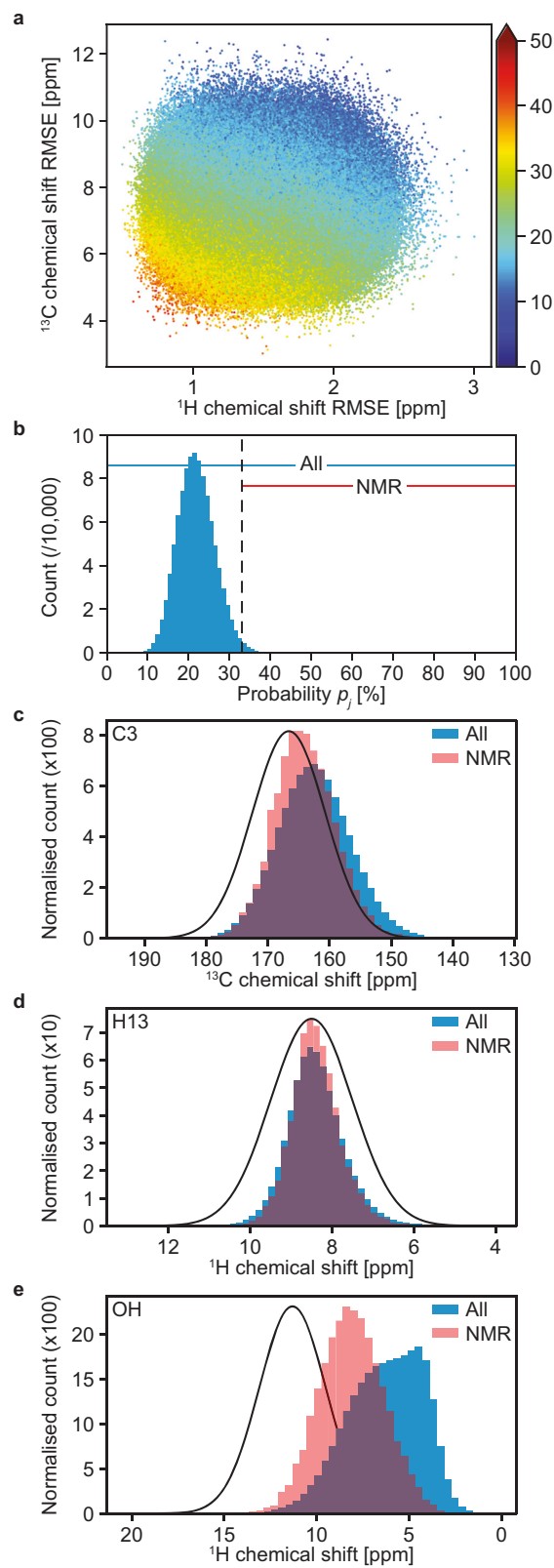

**Fig. 2 | Ensemble structure determination. a** A comparison of $^1$H and $^{13}$C chemical shift RMSEs for each molecule in the MD snapshots, colored according to its probability to be simultaneously compatible with the experimental shift distributions for all assigned atoms (as described by Eqs. 1–4 in the Methods section). **b** Histogram of the probabilities of all molecules in the MD snapshots to be compatible with the experimental shift distributions. The dashed line indicates the probability threshold used to select local molecular environments. The ranges of probabilities included in the whole and the NMR ensembles are indicated above the histogram. Examples of the predicted chemical shift distributions for the (**c**) carbon labeled 3, (**d**) proton labeled 13, and (**e**) OH proton in all molecular environments (blue) in the MD snapshots and in the NMR ensemble (red), compared to the corresponding experimentally measured distributions (black). Equivalent figures for all the other assigned atoms are given in Supplementary Figs. 7–11.

Figure 3 shows the analysis of structural properties in the set of best-match molecular environments, compared to all molecular environments present in all MD snapshots. As seen in Fig. 3a, the selection of local molecular environments compatible with the NMR experiments promotes hydrogen bonds, in particular with the oxygen labeled 3 and the nitrogen labeled c. Accordingly, the proportion of OH protons not forming hydrogen bonds is significantly reduced in the set of selected local molecular environments. Hydrogen bonding to nitrogen was found to generally lead to further deshielding of the OH proton compared to hydrogen bond to oxygen, as seen in Supplementary Fig. 12.

Preferred conformations of AZD4625 can be extracted from the NMR ensemble. Figure 3b shows that the position of the OH proton is generally preferred to be pointing away from the body of the molecule, and that this trend is slightly reinforced in the NMR ensemble. Similarly, the $Z$ conformation of the enone group is found to be preferred, and that preference is retained in the NMR ensemble (Fig. 3c). The conformation yielding dihedral angles between the aromatic planes from −120 to −90° were found to be promoted in the NMR ensemble (Fig. 3d). We note that for this case, five of the eight MD simulations carried out started with a dihedral angle around −90° and three of them started with an angle around 90°, which explains the difference in the height of the distributions for positive and negative values in all molecules from the MD snapshots (more details are given in SI). The chair conformation of the aromatic 6-membered ring was also found to be promoted by the NMR selection of local molecular environments compared to the boat conformation that was also observed in the MD simulations (Fig. 3e).

It is interesting to compare the total radial distribution function $G(r)$ and differential correlation function $D(r)$ obtained from the ensembles before and after the selection of local molecular environments with the functions obtained experimentally by powder X-Ray diffraction (Fig. 4). The MD trajectories were found to accurately reproduce the experimental data, with the largest differences found in the two peaks at 1.4 and 2.4 Å. This can be attributed to differences in bond lengths between the MD simulations and the sample. Importantly, the features at distances above 3 Å are correctly captured by the MD simulation. The selection of local molecular environments was not found to significantly change the similarity between the simulated and experimental $G(r)$ or $D(r)$. This result highlights that the scattering data is unable to sensitively discriminate between ensembles of local molecular environments in the samples studied here.

Figure 5 shows the predicted formation energies of molecules of AZD4625 with their local environment, including the formation energy of the central molecule (as described in the Methods section). This is a measure of the stabilization of the molecules by their environment. The local environments in the NMR ensemble were found to result on average in stabilization of the central molecule as compared to random local molecular environments extracted from the MD simulations, by 8.7 ± 0.7 kJ/mol on average (Fig. 5a). This result suggests that that the selection of molecular environments,

match structures, the overlap with the predicted distribution is not perfect. We attribute this to the significant proportion of OH protons weakly bonded to hydrogen bond acceptors in the MD trajectories (see Supplementary Fig. 12). This effect may also be due to bias in the shift predictions. We also note that importantly the best match selection does not critically depend on any single shift, but is the result of the joint match to all the shifts in the molecule.

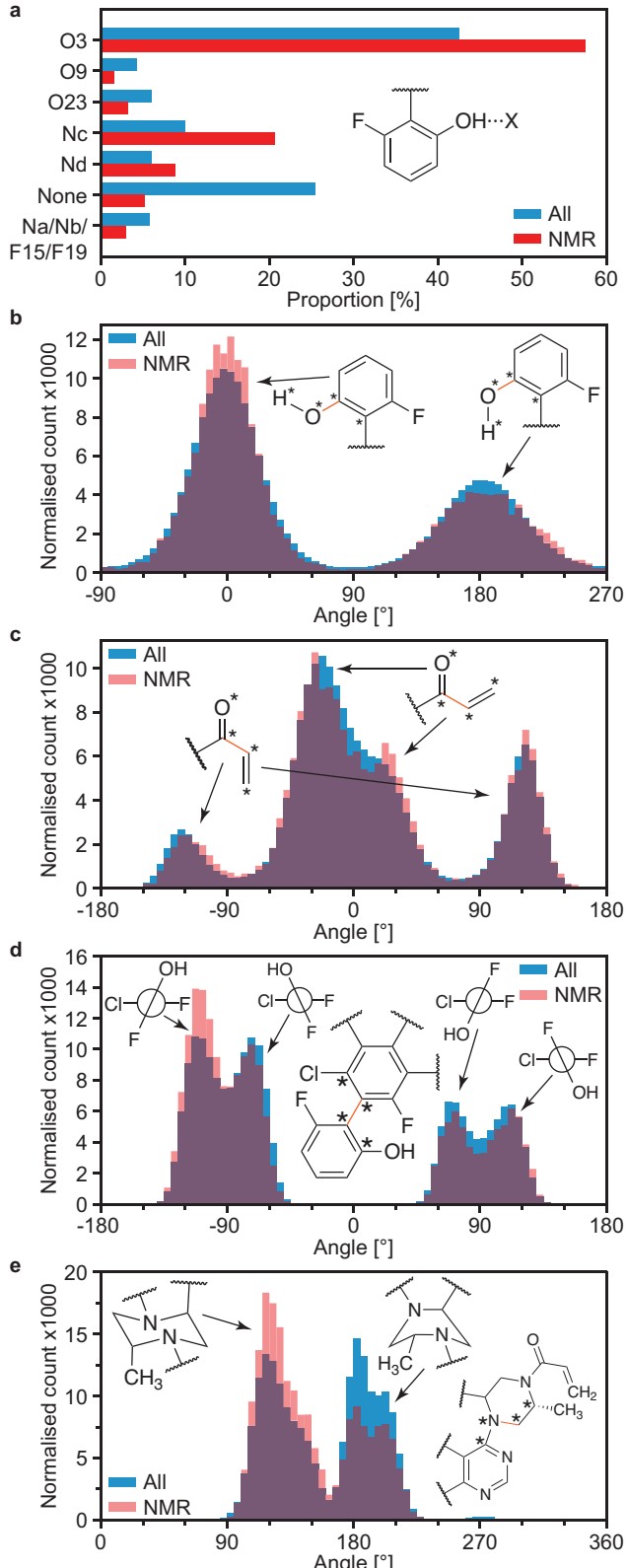

**Fig. 3 | Structural properties of the amorphous form of AZD4625. a** Proportions of different hydrogen bond acceptors bonded to the OH group of AZD4625 in all local molecular environments (blue) and in the NMR ensemble (red). Histogram of dihedral angles for the **b** OH group, **c** enone, **d** aromatic planes, and **e** aliphatic ring in all molecules (blue) and in the NMR ensemble (red). In **b–e** the rotatable bond associated with the dihedral angle is drawn in orange. Stars indicate the atoms used for the computation of the dihedral angle.

based purely on NMR chemical shifts, also led to the selection of energetically favorable local molecular environments. Figure 5b shows that hydrogen bonding of the OH proton of a central molecule to either oxygen labeled 3 or nitrogen labeled d leads to enhanced stabilization of the central molecule by its whole environment. This also corroborates the increase in hydrogen bonds formed with these two atoms in the NMR ensemble of molecular environments discussed above (Fig. 3a).

A set of 20 randomly selected central molecules from the NMR ensemble is shown in Fig. 6a. This highlights the structural flexibility of AZD4625 in the amorphous state. Fig. 6b shows three-dimensional atomic density maps around the OH proton in the NMR (left panel) and the random (middle panel) local molecular environments, as well as the difference between the two atomic density maps (right panel). As expected from Figs. 3a and 5b, hydrogen bonding towards oxygen and nitrogen atoms is promoted by the selection of local molecular environments. This is highlighted by the contours representing nitrogen and oxygen atomic densities in the rightmost panel in Fig. 6b. This suggests that these interactions are critical to stabilizing the structure of amorphous AZD4625. Figure 6c shows similar atomic density maps, aligned around the methyl group of AZD4625. The difference between atomic density maps highlights the preferred conformation of the 6- and 8-membered aliphatic rings.

## Discussion

We have determined the ensemble atomic-level structure of the amorphous form of AZD4625 by combining solid-state NMR experiments with MD simulations and prediction of chemical shifts for over one million AZD4625 molecules in the MD trajectories. Importantly, no crystalline structure of the pure compound has previously been reported.

Local molecular environments compatible with the NMR spectra measured were selected through a general approach that integrates multiple chemical shifts, and includes the spread of chemical shift distributions in the experimental spectra as well as the uncertainty of the chemical shift predictions. We expect that the method presented here can be straightforwardly applied to determine the structure of any molecular solid.

The local atomic environments determined by NMR were found to accurately reproduce the radial distribution function measured for the sample by powder X-Ray diffraction. The NMR ensemble was also found to lead to an overall stabilization of the selected molecules by their environment.

The ensemble of selected local molecular environments highlights key structural properties in the amorphous sample that play a critical role in the structure and stabilization of the material in its amorphous form.

## Methods
### Synthesis
The synthesis of AZD4625 is described in ref. 81. The amorphous AZD4625 solid was precipitated from 2-methyltetrahydrofuran (2-MeTHF) and n-heptane. Crude API was initially dissolved in 2-MeTHF, the solution of which was charged directly to n-heptane at 18 °C. The precipitate was isolated under vacuum and dried from 25–70 °C.

### X-ray diffraction experiments
Synchrotron X-ray PDF data were collected on the I15-1 beamline at Diamond Light Source, UK. Powdered samples were contained within a 1 mm inner diameter polyimide capillary with a 0.025 mm wall thickness and spun perpendicular to the beam during data collection. An empty capillary was also collected for background subtraction. Scattering data were collected at an incident X-ray energy of 76.69 keV with one Perkin Elmer XRD4343CT area detector placed close to the sample (~200 mm) for PDF data and a second Perkin Elmer XRD1611CP3 area

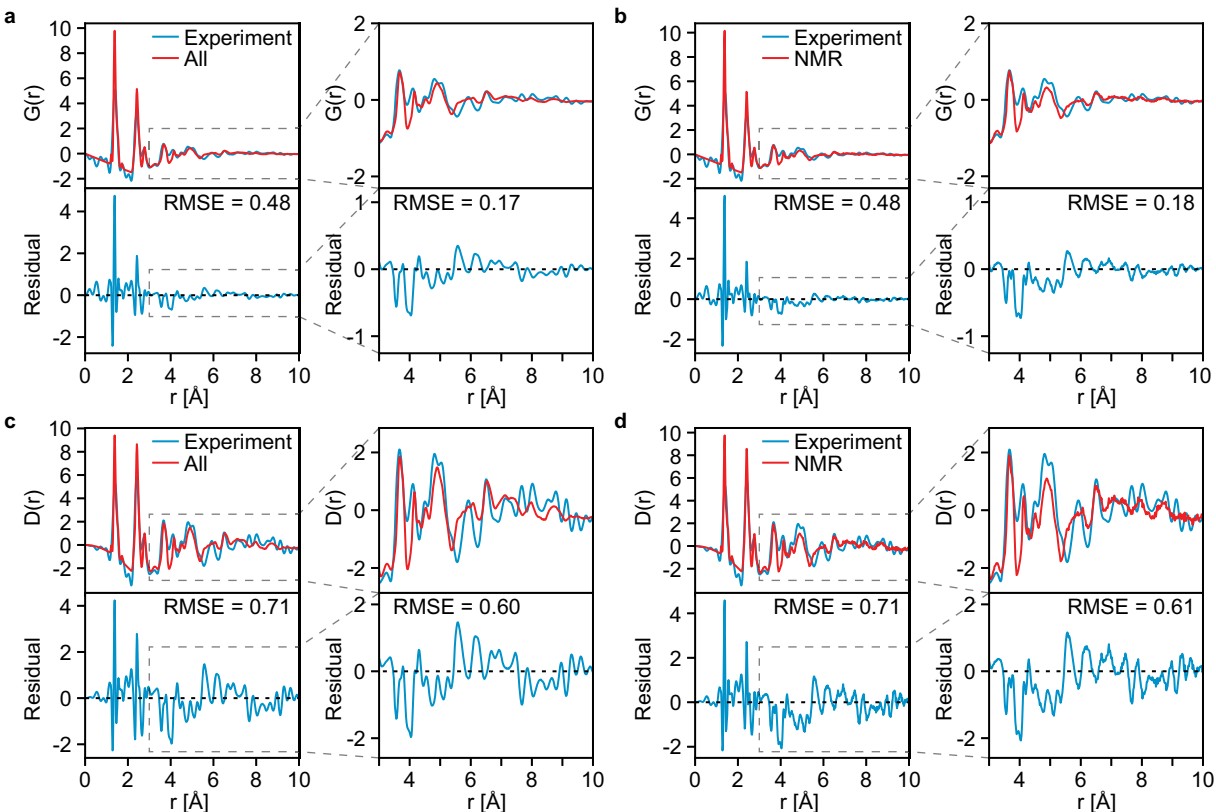

**Fig. 4 | Radial distribution functions.** The total radial distribution function $G(r)$ (**a**, **b**) and the differential correlation function $D(r)$ (**c**, **d**) measured from powder X-ray diffraction (see methods) (blue) and simulated (red) using (**a**, **c**) all molecules and (**b**, **d**) the best match ensemble by NMR. The lower panels show the residual between the experiment and simulations in each case, along with the RMSE obtained. The plots on the right of each panel show the range between 3 and 10 Å, and the RMSE in the corresponding range.

detector placed further from the sample (~850 mm) for higher resolution Bragg data; the precise detector geometries were calculated using DAWN[86] from data collected on a crystalline standard (NIST SRM640c). Total data collection times were 30 min for the PDF data and 2 min for Bragg data. 2D scattering data were corrected for polarization, solid angle, and detector thickness prior to integration to 1D using DAWN[86]. The GudrunX program was then used to perform container background, multiple scattering, Compton scattering, and absorption corrections on data in the range $0.3 \leq Q \leq 26\ Å^{-1}$, prior to Fourier transform to produce the PDF[87].

### NMR experiments

Experiments were carried out using either room temperature ultra-fast MAS rate techniques that enhance [1]H spectral resolution or DNP approaches that enhance the sensitivity of NMR signals. DNP is performed at temperatures of ~100 K and relies on the transfer of high electron spin polarization, typically from exogenously added solutions of organic radicals, to nuclei of interest upon microwave irradiation[48,49,88,89].

The DNP-enhanced NMR experiments were carried out on commercial Bruker Avance Neo NMR spectrometers at a nominal field strength of 9.40 T equipped with either a 264 GHz klystron or a 263 GHz gyrotron microwave source and a 3.2 mm LTMAS DNP probe in a [1]H/[13]C/[15]N configuration which was cooled to about 100 K before sample insertion. The DNP sample was packed into a 3.2 mm sapphire rotor, plugged with a Teflon insert, and topped with a zirconia drive cap. Prior to packing, the powder sample of the amorphous form of AZD4625 was ground by hand in a pestle and mortar and then impregnated[48,49,88,89] with a 20 mM solution of the AMUPol biradical[90] dissolved in a mixture of $H_2O{:}D_2O{:}^{12}C$-glycerol (10:30:60 v/v). A DNP

enhancement of a factor 6–8 was achieved, measured as the ratio of the ([1]H)[13]C cross-polarization (CP) signal intensity between spectra acquired with and without microwaves. While this is a modest enhancement, it was sufficient to enable the acquisition of the natural abundance [13]C-[13]C INADEQUATE experiments described below. DNP spectra were acquired at MAS rates of 8 or 10 kHz.

The room temperature NMR experiments were performed on a dry sample of the powder at a MAS rate of 100 kHz, using a Bruker 0.7 mm room temperature HCN CPMAS probe at a magnetic field of 21.1 T. A States-TPPI acquisition scheme was used to obtain phase-sensitive two-dimensional spectra. The [1]H and [13]C chemical shifts were referenced to literature values. More experimental details and a link to the raw NMR data can be found in the SI.

### Chemical shift assignment

The [1]H and [13]C resonances of the amorphous form of AZD4625 (Fig. 1a) were assigned using one-dimensional [1]H and [13]C MAS NMR experiments, [13]C CPPI spectral editing[91], (Fig. 1b–d, f), in combination with two-dimensional [1]H-[1]H, [13]C-[13]C, and [1]H-[13]C correlation spectra. The [1]H-[1]H DQ/SQ (Fig. 1e) spectrum provides through-space dipolar correlations between protons, the natural abundance DNP-enhanced refocused [13]C-[13]C INADEQUATE[49] (Fig. 1h) provides the covalent connectivities between carbon atoms, and the short- and long-range [1]H-[13]C DNP-Enhanced DUMBO-HETCOR experiments (Fig. 1g, i), provide [1]H-[13]C heteronuclear shift correlations. A DNP-enhanced natural abundance [13]C-[13]C INADEQUATE spectrum recorded for a crystalline form was also used to guide the assignment (Supplementary Fig. 1). The chemical shift assignments obtained from an analysis of these spectra for the [1]H and [13]C nuclei are given in Supplementary Table 6. The chemical shift of C1 was not taken into

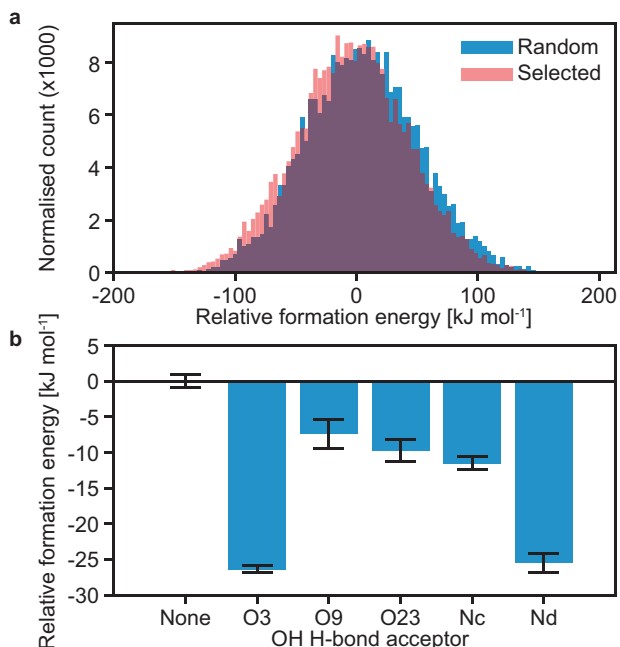

**Fig. 5 | Formation energies. a** Relative formation energies of intermolecular complexes of 8000 randomly selected molecules (blue) and the molecules from the NMR ensemble (red). The zero is set to be the mean formation energy of all intermolecular complexes. **b** Relative formation energies of the local molecular environments in the NMR ensemble for different hydrogen bond acceptors bonded to the OH proton. The zero is set to be the mean formation energy of intermolecular complexes where no hydrogen bonding acceptor is bonded to the OH proton of the central molecule. Formation energies were computed as the difference in energy between a molecular environment (all molecules with at least one atom within 7 Å from any atom of the central molecule) with and without the central molecule, and thus contain both intermolecular interactions and the conformational energy of the central molecule. The error bars shown are the standard error on the mean.

consideration in the subsequent analysis due to high uncertainty in the assignment.

## MD simulation of AZD4625

The amorphous structure of AZD4625 was modeled by carrying out MD simulations with the OPLS4 force-field[92] in Desmond[93,94] on periodic amorphous cells containing 128 molecules. Eight different amorphous cell simulations were generated and evaluated using Materials Studio[95]. After equilibration for 1 ns using the canonical NVT ensemble first at 100 K and then at 298 K followed by 22 ns using the isothermal-isobaric ensemble (NPT) at 298 K and 1 bar, production simulations were carried out for 500 ns using the NPT ensemble at 298 K and 1 bar. Snapshots of each MD simulation were extracted every 100 ps and input directly to ShiftML2[80] for $^1H$ and $^{13}C$ chemical shift predictions. The chemical shielding values were converted to chemical shifts using offsets of 30.78 and 170.04 ppm for $^1H$ and $^{13}C$, respectively. The input files of the MD simulation, extracted MD snapshots, and predicted shifts are given with the raw data. Further information about the MD simulations is given in SI.

## Selection of local molecular environments

Local molecular environments, comprising a central molecule and all other molecules having at least one atom within 7 Å from any atomic site in the central molecule, were extracted from the MD snapshots (1,025,280 environments in total) and selected based on the probability of the molecule at the center of each environment to match the experimental distributions of chemical shifts. Considering one atomic

site $a_i$ in AZD4625, we describe the associated distribution of experimental chemical shifts as a Gaussian function centered on the chemical shift experimentally measured, $\delta_{\exp,a_i}$, and with a width given by the linewidth of the peaks observed in the spectra, $\sigma_{\exp,a_i}$. Based on the measurement of the linewidths in the resolved peaks in the spectra of Fig. 1, here we obtained widths between 2 and 6 ppm for the $^{13}C$ resonances, and 0.6 and 1 ppm for the $^1H$ resonances, except for the OH proton for which we obtained a width of 1.8 ppm. The centers and widths of the experimental chemical shift distributions are given in Supplementary Table 6 and Supplementary Figs. 2–5.

The chemical shift $\delta_{\mathrm{pred},a_i^{(j)}}$ and uncertainty $\sigma_{\mathrm{pred},a_i^{(j)}}$ predicted using ShiftML2 for that atomic site $a_i^{(j)}$ in a molecule $j$ within a given MD snapshot can similarly be described as a Gaussian function centered on the shift prediction and with a width given by the prediction uncertainty. We then define the probability that the computed shift is within the experimental distribution of chemical shift with the two-tailed $p$ value resulting from the $Z$ score computed between the two Gaussians:

$$Z_{a_i^{(j)}} = \frac{\left| \delta_{\exp,a_i} - \delta_{\mathrm{pred},a_i^{(j)}} \right|}{\sqrt{\sigma_{\exp,a_i}^2 + \sigma_{\mathrm{pred},a_i^{(j)}}^2}}. \tag{1}$$

The $p$ value $p_{\mathrm{val}}\left(Z_{a_i^{(j)}}\right)$ thus corresponds to the probability that the computed shift is drawn from the experimental distribution of chemical shift for that atomic site:

$$p_{\mathrm{val}}\left(Z_{a_i^{(j)}}\right) = \sqrt{\frac{2}{\pi}} \cdot \int_{Z_{a_i^{(j)}}}^{\infty} \exp\left(-\frac{x^2}{2}\right) dx. \tag{2}$$

We note that the $p$ value corresponds to the null hypothesis, which is here that the shift is drawn from the experimental distribution. A large $p$ value thus indicates a better correspondence between the predicted shift and experimental distribution. To obtain the probability that the computed shift corresponds to the experimental distribution of shifts, we divide the $p$ value obtained by the prediction uncertainty divided by the first quartile of all predicted uncertainties obtained for that atomic site in all molecules of all MD snapshots, $\sigma_{\mathrm{pred},a_i}^0$, capped to a minimum value of 1. This step was done in order to prevent chemical shifts predicted with very high uncertainty, thus where the shift prediction is unreliable, from being artificially associated with a high probability of corresponding to the experimental distribution.

$$p_{a_i^{(j)}} = \frac{p_{\mathrm{val}}\left(Z_{a_i^{(j)}}\right)}{\max\left(1, \frac{\sigma_{\mathrm{pred},a_i^{(j)}}}{\sigma_{\mathrm{pred},a_i}^0}\right)}. \tag{3}$$

The probability $p_j$ that a given molecular environment $j$ within an MD snapshot corresponds to the experimental spectrum was then evaluated as the geometric mean of the probabilities obtained using Eq. 3 for all protons and carbons in the molecule (except, here, for the protons and carbon labeled 1 in Fig. 1a, due to the high uncertainty in the assignment of that carbon). This probability was computed for all local environments in all MD snapshots:

$$p_j = \left(\prod_i^n p_{a_i^{(j)}}\right)^{\frac{1}{n}} \tag{4}$$

The selection of the ensemble of local molecular environments most compatible with the experimental spectra, that we refer to as the NMR ensemble, was then performed by selecting all environments having an overall probability $p_j$ above 0.33, corresponding to

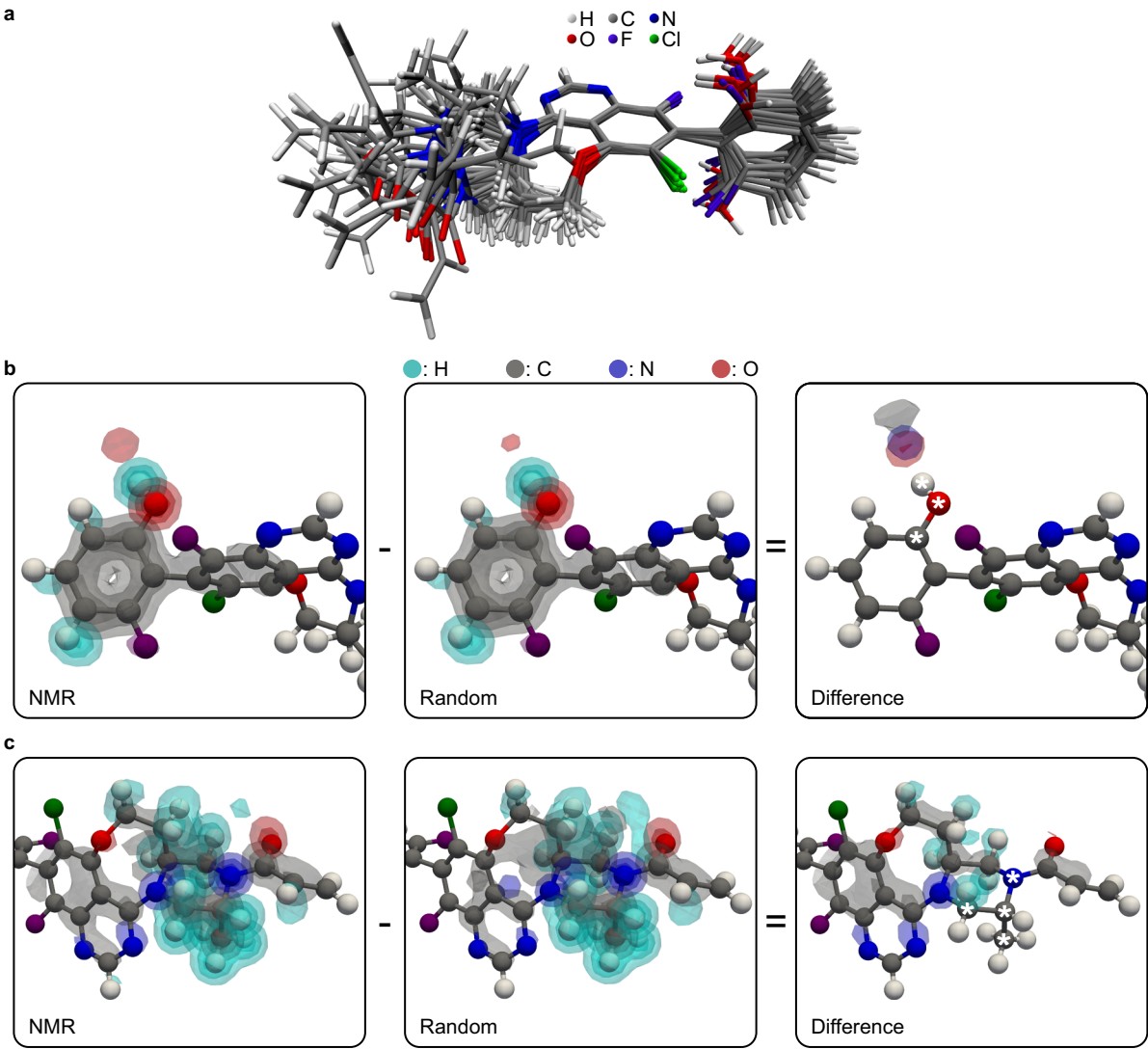

**Fig. 6 | Structures representative of the molecular conformations present in the amorphous form of AZD4625. a** Superposition of 20 molecules of AZD4625 randomly selected from the NMR ensemble. Three-dimensional atomic density maps in NMR-selected and random molecular environments aligned around (**b**) the OH and (**c**) the methyl groups. The difference between the 3D maps for the selected and random molecular environments are shown on the right panels, where the atoms aligned are indicated by asterisks in the difference maps. 3D contours are drawn at levels of 0.2, 0.4, 0.6, and 0.8 for the atomic density maps and 0.05, 0.1, 0.15, and 0.2 for the difference maps. The conformation of the molecule displayed along with the atomic density map was chosen such that the various dihedral angles best correspond to the maxima of the distributions for selected local molecular environments in Fig. 3b–e. The three-dimensional contours in the rightmost panel in **b** and **c** highlight the overall structural features promoted by the NMR-based selection.

about 1% of all local molecular environments present in the MD snapshots (10,107 environments). We note that the cutoff value of 0.33 was chosen as a balance between the maximization of the overlap and minimization of the Jensen-Shannon divergence[96] with the experimental shift distributions, and the selection of large enough ensemble to describe the amorphous compound (see Supplementary Fig. 6).

In addition, 1000 local molecular environments were randomly selected from each MD simulation to construct a random ensemble for comparison with the experimentally determined ensemble.

## Computation of formation energies of local molecular environments

The formation energy of local molecular environments was computed as the energy difference between the environments (all molecules with at least one atom within 7 Å from any atom of the central molecule) with and without the central molecule. This energy thus includes both the intermolecular interactions and conformational energy of the central molecule. The energies were computed using the DFTB-D3H5 semiempirical level of theory using the 3ob-3-1 parameter set and the DFTB+ software version 22.2[97–103]. The computed energies are given with the raw data.

## Identification of hydrogen bonds in local molecular environments

Hydrogen bonds involving the OH proton of the central molecule in each local molecular environment were identified by defining hydrogen bonds as O−H···X motifs (X = O, N) with an O−H−X angle above 130° and H−X distance shorter than 2.5 Å.

## Three-dimensional atomic density maps

The three-dimensional atomic density maps were constructed by aligning the selected and random ensembles of local molecular environments on given atoms in the central molecule. This was done by minimizing the root-mean-square displacement between the positions of the atoms used for the alignment in the central molecule of the

different molecular environments. Three-dimensional atomic density maps were then generated by summing three-dimensional Gaussian functions with a width $\sigma = 0.5\,\text{Å}$ placed at the atomic positions $r_{a_i}$ of the aligned local environments, divided by the number of environments aligned.

$$G(\vec{r}) = \frac{1}{N_{env}} \sum_{i}^{N_{env}} \sum_{a_i \in i} \exp\left(-\frac{|\vec{r} - \vec{r}_{a_i}|^2}{2\sigma^2}\right) \tag{5}$$

Individual atomic density maps were constructed for each element present in the set of aligned environments. The Gaussian functions where not normalized, and this leads to a value of 1 at a given position if an atom of a given element is found at that position in all environments. Each atomic density map was evaluated on a $31 \times 31 \times 31$ cubic grid centered at the aligned atomic sites and with 12 Å sides. This corresponds to a spatial sampling of 0.4 Å.

## Data availability

The NMR raw data are available from the Materialscloud repository https://doi.org/10.24435/materialscloud:gk-51 in JCAMP-DX version 6.0 standard format and original TopSpin format, as well as the input files for the MD simulations, the MD snapshots extracted, formation energies of intermolecular complexes, and all scripts used to perform the data analysis. All data and scripts are available under the license CC-BY-4.0 (Creative Commons Attribution-ShareAlike 4.0 International).

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

## Acknowledgements
This work was supported by AstraZeneca, Swiss National Science Foundation Grant No. 200020_212046, and by the NCCR MARVEL.

## Author contributions
P.M. performed the solid-state NMR experiments. S.N.L. performed the MD simulations. M.C. computed chemical shifts on MD structures, performed the structural analysis, visualization, and scoring, and computed intermolecular complex formation energies. A.C., M.K., J.McC., A.S.A., A.C.P., and S.T.N. prepared and chemically characterized the samples in solid and solution forms. M.C., P.M., S.N.L., S.T.N., J.McC., S.S., and L.E. analyzed the results. S.S and L.E. conceived and supervised the research. M.C., P.M., S.S., and L.E. wrote the manuscript with the contribution of all authors.

## Competing interests
The authors declare no competing interests.
