## [Peer Review File · Nature Communications]

Atomic-Level Structure Determination of Amorphous Molecular Solids by NMRREVIEWER COMMENTS

Reviewer #1 (Remarks to the Author):

This paper employs an innovative combination of solid-state nuclear magnetic resonance (NMR), classical molecular dynamics simulations, and machine-learned NMR chemical shift prediction to investigate the structure of an amorphous pharmaceutical.

The lack of long-range order makes such materials are difficult to characterize structurally. NMR probes the local chemical environment, but interpreting the results is difficult. NMR crystallography employs chemical shift calculations on candidate structures, but that's hard to do for an amorphous system. To make it work, they have to sample large numbers of candidate environments via MD simulations, and then predict chemical shifts cheaply for ~8000 snapshots with their ShiftML machine learning model.

Versions of the individual techniques employed here have all been reported previously.

Rather, the current paper is impressive because of what they accomplish by combining all of these ideas. In addition to gaining insights into the experimental structure, I found the investigation of the similarities and differences between their NMR-derived ensemble and the energy-determined MD one fascinating. They also do a nice job of highlighting the types of conformations and environments of the drug molecule that occur in the system.

Overall, the paper represents a unique and impressive advance and merits publication in Nature Comms. I have only one minor comment:

- I had a hard time understanding the first page of Results (pg 4) until I read the Methods section. I recognize they are adhering to the format of describing the Methods at the end, but minor edits could make the qualitative nature of their probabilistic approach clearer in this section.

Reviewer #2 (Remarks to the Author):

This paper represents significant progress in the characterization of amorphous organic materials from NMR data and as such is worthy of publication in Nature Communications. It does represent a major advance on ref 42 (Cordova, M. et al. Structure determination of an amorphous drug through large-scale NMR predictions. Nat Commun 12, 2964 (2021)) as that only determined the hydrogen-bonding environments of the three N-H protons in a different amorphous drug. This paper shows how the machine learned chemical shifts can enable the determination of the conformation as well as hydrogen bonding in an amorphous drug. My main concern with this paper is that the general title “of amorphous molecular solids” is not supported by one unverified result on a simpler amorphous drug and so the title would be more accurate as “of an amorphous pharmaceutical solid”. This should be published as a communication demonstrating the power of complex multiple NMR experiments combined with theory to elucidate the range of conformations, hydrogen bonding etc seen in an amorphous form of a pharmaceutical, a methodology which I expect will be refined by a wide range of studies. This paper is convincing as being performed on an Astra Zeneca molecule, rather than one chosen for its suitability for the computational modelling, and I expect that further examples refining this approach will follow. Most of my comments are suggestions for clarification to help the reader.

The result of the analysis is only not verified because the experiment that was probably intended to do this (as I cannot think of any other verification approach), the radial distribution functions derived from X-ray diffraction, were unable to discriminate between the ensembles of local molecular environments. This makes the achievement of this paper even more impressive. My initial impression of Fig 4 was that there may have been an error, with identical RMSE's and such similar figures, dominated by the unspecified errors in the MD force-field at short distances. I suggest the diagrams in Fig 4 start at 3 Å and have the current figures as an inset.

The methodology is critically dependent on the whether the machine learned chemical shifts and the MD simulations of the amorphous material fully sample all the conformational and hydrogen-bonding space that is available in the amorphous form. Note that the possibility of polyamorphism, in which the amorphous states differ in the dominant

conformational region or type of hydrogen bonding present, suggests the experimentally as well as computationally, there may be molecule-dependent barriers to fully sampling all possible molecular environments. Hence the starting points for the 8 MD simulations of the amorphous solid, which are limited to 128 molecules. may be critical in determining the coverage of possible configurations. The SI suggests that all 8 amorphous simulations were all generated from one isolated molecule conformation, whereas the text (around Fig 3) suggests that the starting conformations had a dihedral angle between the aromatic planes of around $+90^\circ$ or -90° . (Why does this explain the differences in height in Fig3d?) The SI must contain more details of how the amorphous MD simulations were produced, and why 8 were used. This will be very dependent on the molecule. ADZ4625 has only one hydrogen bond donor OH position, one very flexible dihedral (C15-C16-C18-C19 which gives both R and OH on both sides of the molecule in Fig A), some flexibility in 8 membered ring, boat/chair for 6 membered ring, and enone conformation. This makes for a good test system, and I expect that 8 MD simulations are easily sufficient for ADZ4625's flexibility, but this must be argued so the method can be adapted for other systems.

Fig 2B would be more helpful in interpreting the labels on other figures if the NMR conformations beyond the cutoff were in red and the ranges corresponding to ALL and NMR were marked.

If the formation energies in Fig 5 were calculated using all molecules having at least one atom from any atom in the central molecule, please put this in the figure caption. This was hard to deduce from just looking at the methods section under the heading of formation energies. This is a good test of the preferred conformation/hydrogen bonding, as the average formation energy is reasonably stabilizing.

Fig 6 a is very helpful, and it is useful that B and C are from another angle of view, but they are too small. Does the grey then purplish blob near the H-O proton in the difference show that there is both Nd and O3 as hydrogen bond acceptors? (i.e. would a larger figure make the overlap between different atom colors clearer?)

Are any crystal structures known for ADZ4625? If so, do all the molecular conformation(s)

and hydrogen bonding in the crystals fit within the NMR datasets? Can the crystal conformations be added to Fig 6A in a different color to emphasize this? This would help illustrate the need for characterizing the amorphous state sufficiently to analyze whether it is likely to be stable relative to crystallization. This is correctly identified in the introduction as a reason why this work is so significant.

Reviewer #3 (Remarks to the Author):

Emsley and coworkers have submitted a manuscript detailing their impressive work elucidating the three-dimensional structure of the amorphous form of drug AZD4625. The present work follows on their earlier work developing machine-learned chemical shifts and applying a large-scale ensemble of *in silico* predictions of structures and NMR properties to work out amorphous structures in small organic molecules. I think the present methodology is more elegant and applicable to wide array of small organic molecules, not to mention less computationally intensive.

The techniques presented in the current manuscript instead utilize MD simulations to generate the “NMR ensemble” and local molecular environments, combined with state-of-the-art solid-state NMR and DNP-enhanced NMR to work out the most probabilistic ensemble of structures present in a metastable amorphous phase. Most importantly, in my opinion the authors paid close attention to the critical stabilizing interactions present in amorphous AZD4625. This type of analysis will be of great interest to pharmaceutical scientists working in the field of amorphous solids and amorphous formulation development. It is of great importance to develop stable formulations of amorphous drugs, with or without stabilizing polymeric excipients, and a deeper understanding of stabilizing interactions is warranted. I think the current work takes a big step forward toward this end. Understanding of the interactions made to stabilize a neat API could help lead to improved selection of polymers and other excipients to help stabilize an amorphous drug to prevent crystallization.

I think this work is of considerable novelty and addresses a problem long pondered by many in the field of amorphous solids, particularly in the pharmaceutical arena. The molecule

itself is of structural relevance to the modern pharmaceutical industry, making the work even more relevant to current investigators. All those in the field will find this work of great interest, in my opinion. The data collected are of high quality and provide ample support for the claims made, supported by statistical analysis and controls, for example the randomly selected 1000 structures from the MD ensemble.

The manuscript is well written and support with sufficient data and references, and I fully support publication without further revision.

REVIEWER COMMENTS

Reviewer #1 (Remarks to the Author):

This paper employs an innovative combination of solid-state nuclear magnetic resonance (NMR), classical molecular dynamics simulations, and machine-learned NMR chemical shift prediction to investigate the structure of an amorphous pharmaceutical.

The lack of long-range order makes such materials are difficult to characterize structurally. NMR probes the local chemical environment, but interpreting the results is difficult. NMR crystallography employs chemical shift calculations on candidate structures, but that's hard to do for an amorphous system. To make it work, they have to sample large numbers of candidate environments via MD simulations, and then predict chemical shifts cheaply for ~8000 snapshots with their ShiftML machine learning model.

Versions of the individual techniques employed here have all been reported previously. Rather, the current paper is impressive because of what they accomplish by combining all of these ideas. In addition to gaining insights into the experimental structure, I found the investigation of the similarities and differences between their NMR-derived ensemble and the energy-determined MD one fascinating. They also do a nice job of highlighting the types of conformations and environments of the drug molecule that occur in the system.

We thank the reviewer for their positive feedback.

Overall, the paper represents a unique and impressive advance and merits publication in Nature Comms. I have only one minor comment:

- I had a hard time understanding the first page of Results (pg 4) until I read the Methods section. I recognize they are adhering to the format of describing the Methods at the end, but minor edits could make the qualitative nature of their probabilistic approach clearer in this section.

We have now added short summaries of the methods at the beginning of the results section.

Reviewer #2 (Remarks to the Author):

This paper represents significant progress in the characterization of amorphous organic materials from NMR data and as such is worthy of publication in Nature Communications. It does represent a major advance on ref 42 (Cordova, M. et al. Structure determination of an amorphous drug through large-scale NMR predictions. Nat Commun 12, 2964 (2021)) as that only determined the hydrogen-bonding environments of the three N-H protons in a different amorphous drug. This paper shows how the machine learned chemical shifts can enable the determination of the conformation as well as hydrogen bonding in an amorphous drug. My main concern with this paper is that the general title "of amorphous molecular solids" is not supported by one unverified result on a simpler amorphous drug and so the title would be more accurate as "of an amorphous pharmaceutical solid". This should be published as a communication demonstrating the power of complex multiple NMR experiments combined with theory to elucidate the range of conformations, hydrogen bonding etc seen in an amorphous form of a pharmaceutical, a methodology which I expect will be refined by a wide range of studies. This paper is convincing as being performed on an Astra Zeneca molecule, rather than one chosen for its suitability for the computational modelling, and I expect that further examples refining this approach will follow. Most of my comments are suggestions for clarification to help the reader.

We thank the reviewer for their positive feedback.

In principle, there is no particular feature of the compound studied here that makes it particularly suitable to the analysis performed. We believe that the method presented here is agnostic to the molecule studied, and can be straightforwardly used to determine the structure of other amorphous molecular solids. (We believe that the reviewer comment in the sentence following the suggestion to make the title more specific, concluding with "a methodology which I expect will be refined by a wide range of studies" actually sums this up nicely!) This is now mentioned in the discussion. In light of that, we believe that the current title is accurate.

The result of the analysis is only not verified because the experiment that was probably intended to do this (as I cannot think of any other verification approach), the radial distribution functions derived from X-ray diffraction, were unable to discriminate between the ensembles of local molecular environments. This makes the achievement of this paper even more impressive. My initial impression of Fig 4 was that there may have been an error, with identical RMSE's and such similar figures, dominated by the unspecified errors in the MD force-field at short distances. I suggest the diagrams in Fig 4 start at 3 Å and have the current figures as an inset.

Figure 4 now includes plots starting at 3 Å, along with the corresponding RMSEs.

The methodology is critically dependent on the whether the machine learned chemical shifts and the MD simulations of the amorphous material fully sample all the conformational and hydrogen-bonding space that is available in the amorphous form. Note that the possibility of polyamorphism, in which the amorphous states differ in the dominant conformational region or type of hydrogen bonding present, suggests the experimentally as well as computationally, there may be molecule-dependent barriers to fully sampling all possible molecular environments. Hence the starting points for the 8 MD simulations of the amorphous solid, which are limited to 128 molecules. may be critical in determining the coverage of possible configurations. The SI suggests that all 8 amorphous simulations were all generated from one isolated molecule conformation, whereas the text (around Fig 3) suggests that the starting conformations had a dihedral angle between the aromatic planes of around +90° or -90°. (Why does this explain the differences in height in Fig3d?) The SI must contain more details of how the amorphous MD simulations were produced, and why 8 were used. This will be very dependent on the molecule. ADZ4625 has only one hydrogen bond donor OH position, one very flexible dihedral (C15-C16-C18-C19 which gives both R and OH on both sides of the molecule in Fig A), some flexibility in 8 membered ring, boat/chair for 6 membered ring, and enone conformation. This makes for a good test system, and I expect that 8 MD simulations are easily sufficient for ADZ4625's flexibility, but this must be argued so the method can be adapted for other systems.

We agree that it is indeed important to fully sample the conformational and noncovalent interaction space in order to obtain a comprehensive analysis of the system. (This problem is not unique to our approach, but is a factor in any MD driven structure determination method).

To attempt to fully sample the space, we used a single conformation of the molecule with different arrangements of the 128 molecules in cells as a starting point for the 8 MD simulations, except for the orientation of the aromatic ring containing the OH group. In order to randomize the conformations, the simulations were equilibrated for 23 ns before the production runs, and we only considered the last 100 ns (out of a total of 500 ns) of the production simulations to ensure complete equilibration of the systems before the analysis. Since no inversion of the aromatic ring containing the OH group was observed in the MD simulations, we performed 5 MD simulations with the starting angle between the aromatic planes around -90° and 3 with a starting angle around 90°. There is thus a 5:3 ratio between the number of conformations with negative and positive angles.

More details have now been added to the SI, as requested.

Fig 2B would be more helpful in interpreting the labels on other figures if the NMR conformations beyond the cutoff were in red and the ranges corresponding to ALL and NMR were marked.

We added the ranges corresponding to All and NMR in the figure. Since the conformations beyond the cutoff are both in the All and NMR ensembles, we did not change their color.

If the formation energies in Fig 5 were calculated using all molecules having at least one atom from any atom in the central molecule, please put this in the figure caption. This was hard to deduce from just looking at the methods section under the heading of formation energies. This is a good test of the preferred conformation/hydrogen bonding, as the average formation energy is reasonably stabilizing.

Yes, the energies were computed using all molecules with at least one atom within 7 Å from any atom of the central molecule. This is now mentioned in the caption of figure 5 and in the methods section.

Fig 6 a is very helpful, and it is useful that B and C are from another angle of view, but they are too small. Does the grey then purplish blob near the H-O proton in the difference show that there is both Nd and O3 as hydrogen bond acceptors? (i.e. would a larger figure make the overlap between different atom colors clearer?)

Indeed there are two “blobs” superimposed in front of the OH proton: a red one and a blue one, corresponding to enhanced H-bonding to oxygen and nitrogen atoms in the NMR ensemble. We have made Figure 6 larger in order to better visualize the structure identified.

Are any crystal structures known for ADZ4625? If so, do all the molecular conformation(s) and hydrogen bonding in the crystals fit within the NMR datasets? Can the crystal conformations be added to Fig 6A in a different color to emphasize this? This would help illustrate the need for characterizing the amorphous state sufficiently to analyze whether it is likely to be stable relative to crystallization. This is correctly identified in the introduction as a reason why this work is so significant.

There is no pure crystalline phase of AZD4625 reported so far. (Which we believe increases the importance of the work here, since the ensemble determined here is the only structure so far for the pure drug). This is now mentioned in the text.

Reviewer #3 (Remarks to the Author):

Emsley and coworkers have submitted a manuscript detailing their impressive work elucidating the three-dimensional structure of the amorphous form of drug AZD4625. The present work follows on their earlier work developing machine-learned chemical shifts and applying a large-scale ensemble of in silico predictions of structures and NMR properties to work out amorphous structures in small organic molecules. I think the present methodology is more elegant and applicable to wide array of small organic molecules, not to mention less computationally intensive.

The techniques presented in the current manuscript instead utilize MD simulations to generate the “NMR ensemble” and local molecular environments, combined with state-of-the-art solid-state NMR and DNP-enhanced NMR to work out the most probabilistic ensemble of structures present in a metastable amorphous phase. Most importantly, in my opinion the authors paid close attention to the critical stabilizing interactions present in amorphous AZD4625. This type of analysis will be of great interest to pharmaceutical scientists working in the field of amorphous solids and amorphous formulation development. It is of great importance to develop stable formulations of amorphous drugs, with or without stabilizing polymeric excipients, and a deeper understanding of stabilizing interactions is warranted. I think the current work takes a big step forward toward this end. Understanding of the interactions made to stabilize a neat API could help lead to improved selection of polymers and other excipients to help stabilize an amorphous drug to prevent crystallization.

I think this work is of considerable novelty and addresses a problem long pondered by many in the field of amorphous solids, particularly in the pharmaceutical arena. The molecule itself is of structural relevance to the modern pharmaceutical industry, making the work even more relevant to current investigators. All those in the field will find this work of great interest, in my opinion. The data collected are of high quality and provide ample support for the claims made, supported by statistical analysis and controls, for example the randomly selected 1000 structures from the MD ensemble.

The manuscript is well written and support with sufficient data and references, and I fully support publication without further revision.

We thank the reviewer for their very positive feedback.

REVIEWERS' COMMENTS

Reviewer #1 (Remarks to the Author):

The authors have addressed my minor comments, and it is now suitable for publication.

Reviewer #2 (Remarks to the Author):

The manuscript has been considerably improved by the revision. The description in the SI of the choice of starting points for the 8 MD simulations now suggests that the lack of change in the OH conformation was noted in first few simulations and so changed in the starting point for others. This difficulty in the MD sampling is part of my concern about the generality of the methodology, along with the limitations of semi-empirical calculations of cluster energies. However, I am happy to recommend this manuscript for publication.

REVIEWERS' COMMENTS

Reviewer #1 (Remarks to the Author):

The authors have addressed my minor comments, and it is now suitable for publication.

We thank the reviewer for their positive feedback.

Reviewer #2 (Remarks to the Author):

The manuscript has been considerably improved by the revision. The description in the SI of the choice of starting points for the 8 MD simulations now suggests that the lack of change in the OH conformation was noted in first few simulations and so changed in the starting point for others. This difficulty in the MD sampling is part of my concern about the generality of the methodology, along with the limitations of semi-empirical calculations of cluster energies. However, I am happy to recommend this manuscript for publication.

We thank the reviewer for their positive feedback.